# Expression of HOXB7 in the Lung of Patients with Idiopathic Pulmonary Fibrosis: A Proof-of-Concept Study

**DOI:** 10.3390/biomedicines12061321

**Published:** 2024-06-13

**Authors:** Anna Valeria Samarelli, Roberto Tonelli, Giulia Raineri, Ilenia Mastrolia, Matteo Costantini, Luca Fabbiani, Virginia Catani, Tiziana Petrachi, Giulia Bruzzi, Dario Andrisani, Filippo Gozzi, Alessandro Marchioni, Valentina Masciale, Beatrice Aramini, Valentina Ruggieri, Giulia Grisendi, Massimo Dominici, Stefania Cerri, Enrico Clini

**Affiliations:** 1Respiratory Diseases Unit, University Hospital of Modena, 41124 Modena, Italy; annavaleria.samarelli@unimore.it (A.V.S.); giuliaraineri@unimore.it (G.R.); giulibru92@gmail.com (G.B.); andrisanidario@gmail.com (D.A.); fillo.gzz@gmail.com (F.G.); marchioni.alessandro@unimore.it (A.M.); valentina.ruggieri@unimore.it (V.R.); stefania.cerri@unimore.it (S.C.); enrico.clini@unimore.it (E.C.); 2Laboratory of Cellular Therapies and Respiratory Medicine, Department of Medical and Surgical Sciences for Children & Adults, University Hospital of Modena and Reggio Emilia, 41124 Modena, Italy; 3Laboratory of Cellular Therapy, Department of Medical and Surgical Sciences, University of Modena and Reggio Emilia, 41124 Modena, Italy; ilenia.mastrolia@unimore.it (I.M.); virginia.catani@unimore.it (V.C.); valentina.masciale@unimore.it (V.M.); giulia.grisendi@unimore.it (G.G.); massimo.dominici@unimore.it (M.D.); 4Pathology Unit, University Hospital of Modena, 41124 Modena, Italy; costantini.matteo@aou.mo.it (M.C.); luca.fabbiani@unimore.it (L.F.); 5Clinical and Experimental Medicine PhD Program, University of Modena Reggio Emilia, 41125 Modena, Italy; 6Technopole “Mario Veronesi”, Via 29 Maggio 6, 41037 Mirandola, Italy; tiziana.petrachi@tpm.bio; 7Division of Thoracic Surgery, Department of Medical and Surgical Sciences, DIMEC of the Alma Mater Studiorum, University of Bologna, GB Morgagni-L Pierantoni Hospital, 47121 Forlì, Italy; beatrice.aramini2@unibo.it; 8Department of Oncology and Hematology, University Hospital of Modena, 41124 Modena, Italy

**Keywords:** interstitial lung disease, idiopathic pulmonary fibrosis, lung fibrosis, homeobox, HOXB7, biomarkers

## Abstract

Background: The molecular pathways involved in the onset and progression of idiopathic pulmonary fibrosis (IPF) still need to be fully clarified as some are shared with lung cancer development. HOXB7, a member of the homeobox (*Hox*) gene family, has been found involved in various cancers. Methods: Immunohistochemical (IHC) analysis was run on lung tissue samples from surgical lung biopsy (SLB) of 19 patients with IPF, retrospectively selected from the IPF database of the University Hospital of Modena. HOXB7 expression was analyzed and compared with that of five patients with no evidence of pulmonary fibrosis as controls. Results: The semi-quantitative analysis of IHC showed that HOXB7 protein expression was higher in IPF patients compared to controls (difference between means = 6.2 ± 2.37, *p* = 0.0157). Further, HOXB7 expression was higher in IPF patients with a higher extent of fibrosis (50–75%)—measured with high-resolution computer tomography—compared to those with a lower extent (0–25%) (difference between means = 25.74 ± 6.72, *p* = 0.004). Conclusions: The expression of HOXB7 is higher in the lung of IPF patients compared to controls, and was represented in different cellular compartments within the lung niche. Further investigations are needed to clarify its role in the pathogenesis and progression of IPF.

## 1. Introduction

Idiopathic pulmonary fibrosis (IPF) is a chronic and progressive interstitial lung disease (ILD) characterized by poor prognosis and severely impaired quality of life [1]. Despite the approval of two drugs (i.e., nintedanib and pirfenidone) that proved effective in slowing down the decline of respiratory function of IPF patients, there is no curative treatment currently available [2,3]. Several key processes are described that contribute to the development and progression of the disease. These include dysregulated wound healing in response to cyclic epithelial lung injury, fibroblast activation and proliferation with excessive extracellular matrix deposition (ECM), stem cell senescence, and genetic predisposition [4]. Further, immune cells seem to play a role as the activation of adaptive and innate immunity dysregulates the expression level of pro- and anti-inflammatory cytokines [5]. Nevertheless, as the precise mechanisms of IPF pathogenesis are not fully clarified, understanding the critical pathways that set off the disease and trigger its progression still represents an unmet biological need. The homeobox (*Hox*) genes constitute a family of transcription factors governing cell adhesion and motility, receptor signaling, differentiation, apoptosis, and stem cell self-renewal [6]. *Hoxb7*, a member of the homeobox gene family, has been demonstrated to play a crucial role in regulating proliferation, motility, and angiogenesis in various solid and non-solid neoplastic diseases, including leukemia [7], melanoma [8], breast [9,10], colorectal [11], head and neck [12], pancreatic [13], oral [14], and lung cancer (LC). LC has been frequently observed in IPF, particularly in current smoking patients exhibiting a rapid decline in forced vital capacity (FVC) [15,16]. For this purpose, several recent studies have identified common molecular pathways between IPF and lung cancer whose better knowledge could result in novel therapeutic strategies and optimal management of patients with both diseases. Among common cellular and molecular mechanisms that might predispose patients to the onset and development of IPF and LC, the most representative ones are the activation and proliferation of both myofibroblasts (IPF) and cancer-associated fibroblasts (CAFs), and the alteration of growth factors’ expression levels [17,18]. In particular, myofibroblasts are considered the cellular key players in IPF since the proliferation and activation under profibrotic stimuli trigger the secretion and deposition of ECM proteins, which are responsible for the onset and the progression of fibrosis [19,20]. CAFs, which predominantly constitute the tumour stroma and contribute to the biology of the tumour, are actively involved in the mechanisms of proliferation, invasion, inflammation, angiogenesis, and metastasis that take place in the tumour microenvironment (TME). Consequently, given the commonly shared molecular mechanisms between IPF and LC onset and progression and the pivotal role of HOXB7 in the main processes of tumour progression, we aimed to investigate the expression and the cellular localization of HOXB7 in lung tissue of IPF patients as well as exploring its relationship with disease severity.

## 2. Materials and Methods

### 2.1. Study Design, Setting, and Patient Identification

This study was carried out at the Center for Rare Lung Diseases of the University Hospital of Modena (Italy) and conducted in accordance with the Ethics Committee “Area Vasta Emilia Nord” approval (registered protocol number 557/2019/SPER/AOUMO). Informed consent—to participate in the study and to allow their clinical data to be analyzed and published—was obtained from participants. IPF patients with surgical lung biopsy (SLB) referred to the Center for Rare Lung Diseases of the University Hospital of Modena between December 2011 and January 2020 were retrospectively considered eligible for enrollment. Inclusion criteria were as follows: age >18 years; histologically confirmed diagnosis of IPF according to ATS/ERS/JRS/ALAT guidelines [21,22]. Exclusion criteria were a diagnosis of combined pulmonary fibrosis and emphysema (CPFE); evidence of lung cancer at the time of SLB; missing core data at record analysis. Patients referred to the Respiratory Disease Unit of the University Hospital of Modena (Italy) who underwent SLB and/or surgical lung lobectomy within the diagnostic and therapeutic workup of lung cancer served as eligible controls considering the portion of the lung parenchyma distal to the tumour nodule and the peritumour portion. Thus, the controls were represented by portions of lung parenchyma, histologically conserved, from patients without fibrosing pathology. Then, according to these criteria, of the twenty histological samples of patients with idiopathic pulmonary fibrosis, one patient was excluded because they were subsequently diagnosed with pulmonary fibrosis secondary to scleroderma and were not idiopathic. Of the twenty available controls, only five met the eligibility criteria for this study, selected as they had preserved and normal pulmonary alveolus architecture in comparison with the fibrotic patients.

#### 2.1.1. Sample Selection and Immunohistochemical Analysis

Formalin-fixed and paraffin-embedded (FFPE) samples of IPF patients and controls were retrieved from the archive of the Institute of Pathology of the University Hospital of Modena. For each IPF patient, a preliminary screening of the most representative portion of fibrotic tissue was retrospectively performed on hematoxylin and eosin (H&E, Roche, Basel, Switzerland)-stained tissue prepared for routine IPF diagnostic examination of lung biopsy and collected in the archive of the Pathological Anatomy Unit of the University Hospital of Modena. The corresponding FFPE IPF samples were then cut to obtain 5 μm thick sections. For IHC analysis, the following antibodies were employed: rabbit polyclonal antiHOXB7 [H00003217-D01P] (Abnova, Taipei City, Taiwan; dilution 1:50); rabbit polyclonal anti-SFPTC-1 (Abcam ab90716, Cambridge, UK; dilution 1:1000); rabbit monoclonal anti-CD90 [EPR3133] (Abcam, ab133350; dilution 1:500); rabbit monoclonal anti-gp36 (Abcam, ab236529; 1:2000); mouse monoclonal anti-CD163 (Millipore-Sigma, Burlington, MA, USA; MRQ-26); α-SMA (Millipore-Sigma, EP188). IHC reaction was performed using a DAB Ultraview Universal Detection Kit and the BenchMark XT fully automated IHC slide staining instrument (Roche, Basel, Switzerland). Negative controls omitting primary antibodies were run concurrently. For controls, the sample portion distal to the peritumoural region that showed no evidence of interstitial lung disease was considered for analysis.

#### 2.1.2. Semi-Quantitative Analysis of HOXB7 Expression

The HOXB7 expression was evaluated by double visual analysis with the Zeiss Axioscope (Zeiss, Oberkochen, Baden-Württemberg, Germany). A total of 20 micrographs were randomly acquired at 20× magnification to cover the whole tissue slide surface. The percentage of the positive-stained area for both HOXB7 towards the 3,3′-Diaminobenzidine (DAB) signal and the nuclei toward hematoxylin were calculated for each image using the Color Deconvolution 2 plug-in by ImageJ 1.53t (National Institutes of Health, USA) [23]. The input to the plug-in was a background-corrected bright-field image in 24-bit RGB format of histo/cytology specimens stained with light-absorbing dyes and assumed to have negligible light scattering.

### 2.2. Clinical Data Collection and Case Stratification

Medical reports, electronic charts, and available clinical and physiological datasets were investigated to collect data on demographics, clinical characteristics, pulmonary function tests (PFTs), and radiological features. Cases were then stratified according to the GAP index and stages [24]. Further, for each case, the High Resolution Computed Tomography (HRCT) scan was performed within 6 months before SLB was selected. An expert pneumologist experienced in HRCT interpretation was asked to review images and to quantify the extent of fibrosis into 3 categories (0–25%, 25–50%, and 50–75%) that served as further case stratification.

### 2.3. Analysis Plan and Statistics

The primary explorative aim of this study was to evaluate the expression of HOXB7 as assessed by IHC analysis in the FFPE lung samples of patients with IPF as compared to controls. For this purpose, Student’s t-test was used. To explore the localization of HOXB7 within cellular compartments (i.e., nucleus vs. cytosol), Pearson’s correlation coefficient was sought to assess the correlation between HOXB7 expression and hematoxylin staining. A one-way ANOVA test for multi-comparison analyses among groups (Tukey’s multiple comparisons test) was used to assess the expression of HOXB7 according to GAP index and disease stage. Spearman’s nonlinear correlation was used to explore the correlation between HOXB7 expression and PFTs and the fibrosis extent as expressed by radiological involvement on HRCT. Data were displayed as median and IQR (interquartile range) for continuous variables and numbers and percentages for dichotomous variables.

To further investigate the localization of HOXB7 in the lung parenchyma of IPF patients and controls, an IHC analysis for specific markers of the lung niche using serial tissue slides was performed. For this purpose, we selected the IPF patient with the most extensive staining for HOXB7 from immunohistochemistry analyses to investigate a potential correlation of the expression profile of HOXB7 with the different lung niche markers in all serial tissue slides. In this line, thymocyte differentiation antigen-1 (Thy1, CD90) was used both as a surface marker of stromal cells (i.e., lung fibroblasts CD90^+^) [25] and lung resident mesenchymal stromal cells (LR-MSCs) [26]; CD163 for the detection of alveolar macrophages (AMs); SFTPC gene encoding for surfactant protein-C (SP-C) as a marker of type II alveolar epithelial cell (AECII) [27]; GP36 or podoplanin as a marker of type I alveolar epithelial cell (AECI) [28,29]. Furthermore, we used the Alpha Smooth Muscle Actin (α-SMA) as a marker of the acquired mesenchymal phenotype [30]. Then, the positive marker area, as a percentage, was calculated for each marker mentioned above and according to the protocol described in Section 2.1.2.

Statistical analysis was performed using GraphPad Prism version 9.0 (GraphPad Software, Inc., La Jolla, Ca, USA) unless otherwise indicated. A *p*-value < 0.05 was considered statistically significant.

## 3. Results

### 3.1. Study Population

A total of 19 patients with IPF were enrolled according to inclusion criteria while 5 patients with lung cancer diagnosis served as controls since the normal lung parenchyma far from lung cancer, without fibrosing patterns, was analyzed. A summary of the demographics, clinical, and functional features of the patient cohorts is illustrated in Table 1. The median patient age at the time of biopsy was 69 years (range 50–79), and the majority of patients were male (68.4%).

### 3.2. HOXB7 Expression in the Lung of IPF Patients and Localization in Cellular Compartments

We collected histological data on the expression of the HOXB7 protein in the lung parenchyma of enrolled patients, quantifying both the HOXB7 expression level and hematoxylin signal, resulting in the percentage of the stained area.

Here, HOXB7 was expressed in the nuclei (arrows) and cytosol of both normal (Figure 1A) and fibrotic lung parenchyma (Figure 1B–D). As shown in Figure 1E, the percentage of the stained area related to hematoxylin was significantly higher in IPF patients compared to controls (*p* = 0.0454), suggesting a proliferative phase of the fibrotic lung where different cell populations contribute to the onset and progression of the disease. Then, HOXB7 expression was significantly higher in IPF patients compared to controls (*p* = 0.0157, Figure 1F). A medium positive linear correlation was found between the HOXB7 protein expression and hematoxylin (r = 0.53 95% CI [0.116–1.114], *p* = 0.0158, Appendix A) in the lung parenchyma of patients with IPF suggesting partial localization of the HOXB7 protein in the nuclear compartment of proliferating cells during the progression of the disease. The weak linear correlation between the HOXB7 protein expression and hematoxylin (r < 0.1 *p* = 0.956, ns, Appendix A) in control patients suggests that in the lung parenchyma of control patients, HOXB7 is not primarily present in the nucleus of normal lung alveolar cells. This result shows that HOXB7 is distributed in different cellular compartments in the lung parenchyma of fibrotic patients.

#### 3.2.1. Correlation between HOXB7 and Clinical and Respiratory Functional Status

Given the significant increase in the expression level of the HOXB7 protein in IPF patients compared to controls, we assessed if there was a correlation between HOXB7 expression level and clinicopathological variables such as GAP index. We observed that the expression of HOXB7 was not different between the GAP index categories as assessed by a multi-comparison one-way ANOVA (Figure 2A). Similarly, the expression level of HOXB7 did not significantly change between stages I and II (Figure 2B). As shown in Figure 2C,D, a negative correlation was detected between HOXB7 expression and both FVC (r = −0.304, *p* = 0.196) and DLCO (r = −0.202, *p* = 0.392), although these were not statistically significant.

#### 3.2.2. Correlation between HOXB7 and Radiological and Histological Involvement

Given the importance of the radiological pattern in the diagnosis of patients with IPF, we assessed whether a correlation between HOXB7 expression and the fibrotic radiological score from HRCT analysis existed. Specifically, the fibrotic radiological score was defined by three subgroups that expressed the fibrotic extent as a percentage (0–25%, 25–50%, and 50–75%). Here, we observed that HOXB7 expression was higher in the lung of IPF patients with 50–75% extent of fibrosis on HRCT (median: 47.19) compared to the other subgroups (25–50% median: 21.56 *p*-value: 0.0197, 0–25% 22.73 ** *p*-value: 0.0038) (Figure 2E). This could suggest that the increased expression of HOXB7 is indicative of a more severe radiological pattern and, therefore, may be related to a higher degree of fibrosis.

### 3.3. HOXB7 Localization in Lung Niche of IPF Patients

To further investigate the expression profile of HOXB7 in the lung parenchyma of IPF patients and controls, we selected the IPF patient with the highest stained area for HOXB7 (Figure 1D–F, score: 50.8) to perform serial IHC analysis by using both the HOXB7 antibody and those specific for the lung niche (see Section 2.3).

As shown in Figure 3A, we observed a similarity among the positive area for HOXB7 (Figure 3A, arrows), SP-C marker (Figure 3B, arrows), GP-36 (Figure 3D, arrows), and CD163 (Figure 3F, arrows). Conversely, we found few stromal cells (lung fibroblasts and mesenchymal stromal cells) and cells acquiring mesenchymal phenotype that might be positive for HOXB7, comparing the serial section. Then, we selected control patient#2 to further investigate HOXB7 expression within the lung niche. We found that HOXB7 generally was under-expressed in normal alveoli, showing few areas of histological similarity with SP-C, CD90, and GP-36 (arrows in Figure 4B, Figure 4C, Figure 4D, respectively). Then, the semi-quantitative analysis of the stained area for HOXB7 and the markers of the lung niche of IPF patients showed that the HOXB7 stained area was higher in IPF patients compared to control patients, while the GP-36 stained area was increased in normal lung parenchyma, probably due to the pathobiology of idiopathic pulmonary fibrosis involving AECI depletion (Appendix A). The expression levels of both SFTPC and CD163 were similar between the IPF patients and controls, while α-SMA was highly expressed in patients with IPF compared to the controls.

## 4. Discussion

To the best of our knowledge, this is the first study that provides evidence that HOXB7 protein expression is higher in the lung of patients with IPF compared with the lung of patients with no signs of lung fibrosis that represented our controls. Thus far, very few studies have explored the expression of HOXB7 in lung disease, and none of them have focused on the onset and development of lung fibrosis [31,32,33]. Here, we found a significant expression of HOXB7 in the lung parenchyma of IPF patients based on IHC analysis. Despite the relatively small cohort of patients with IPF, our study population was highly representative of the IPF patient population in terms of clinical and functional features.

HOXB7 per se is known to contribute to cancer progression, promoting epithelial-to-mesenchymal transition [34], anticancer drug resistance, and angiogenesis. As such, the amplification and the overexpression of HOXB7 have been found to correlate with poor prognosis in various cancers such as gastric [35], pancreatic [13], and lung cancer [31]. In our series of patients with IPF, we have observed that HOXB7 does not appear to correlate with increased clinical severity of patients with IPF in terms of lung function decline (FVC and DLCO); however, we have quite clearly shown that HOXB7 expression may be correlated with disease extension, as evidenced in the lung scan imaging.

It might be possible that the lack of correlation between HOXB7 expression and lung function derived from the low number of patients included in the analysis and the clinical stage subgroup representation (see Table 1). Nonetheless, it may be possible that the higher cellular proliferation and representation in IPF, mirrored by a higher amount of cell nuclei, explain the results of the HOXB7 positivity. This in turn might drive fibrotic progression leading to a greater extent in the lung parenchyma of patients with IPF compared with controls. Given these premises, a potential association between HOXB7 and IPF severity/progression might be therefore hypothesized. The serial IHC analysis with the specific antibodies of the lung niche (Figure 3 and Figure 4) showed that SFTPC-positive AECII distribute differently in the pulmonary alveolus of the fibrotic patient (Figure 3B) compared to the controls (Figure 4B), occupying the lumen of the pulmonary interstitium, which is normally covered by AECI. It was in this histological region, characterised by AECII that were probably not able to transdifferentiate into AECI—according to the pathobiogenesis of the IPF causing the depletion of AECI and potential AECII hyperplasia—that we observed the most similarity with the staining areas positive for HOXB7. Thus, we can speculate that HOXB7 may contribute to the epithelial barrier dysfunction, involving the depletion of AECI and the hyperplasia of AECII together with the action of proinflammatory cytokines TGF-βTNF-α, IL-1β and reactive oxygen species (ROS) [36]. Furthermore, a potential expression of HOXB7 in alveolar macrophages (AM) could contribute to the existing cross-talk between alveolar macrophages and lung fibroblasts. In particular, it has been found that macrophages might influence the gene expression of fibroblasts according to their activation state, increasing the expression of collagens, αSMA, and TGFβ [37], as well as ECM synthesis and fibroblast proliferation [38]. As such, this cellular involvement and biological action driven by HOXB7 is likely to promote the profibrotic pathway, which is the hallmark of IPF.

### Further Perspectives

Although IPF has still unknown aetiology, there are several risk factors (i.e., cigarette smoking and environmental risk factors) increasing the risk of disease onset and progression. Patients with IPF have also an increased risk of developing lung cancer throughout the natural course of the disease, particularly non-small-cell lung cancer [39]. Moreover, IPF and lung cancer share several cellular and molecular processes responsible for the progression of both diseases such as fibroblast transition, proliferation, and activation, endoplasmic reticulum stress, oxidative stress, and both genetic and epigenetic markers that might predispose IPF patients to lung cancer development [18]. HOXB7 might be involved as part of a common molecular mechanism that can lead from the development and progression of IPF to lung cancer; therefore, a more thorough investigation of these molecular mechanisms could be of critical importance in stratifying lung cancer risk among patients with IPF.

## 5. Conclusions

Given its main limitation, e.g., the small number of patients included, the present study was able to show that HOXB7—a member of the homeobox gene family involved in regulating the proliferation, motility, and angiogenesis of various solid and non-solid neoplastic diseases—is highly represented in the lung tissue of patients with IPF, and seems to be associated with the extension of disease within this organ. We were also able to demonstrate that HOXB7 might be involved both in the proliferative phase of fibrotic disease and in alveolar epithelium dysfunction since its histological expression profile was associated with an aberrant distribution of alveolar type II cells and a depletion of alveolar type I cells, associated with the pathogenesis of IPF. Therefore, it is likely that HOXB7 may act as a biomarker of disease progression.

This section is not mandatory but may be added if there are patents resulting from the work reported in this manuscript.

## Figures and Tables

**Figure 1 biomedicines-12-01321-f001:**
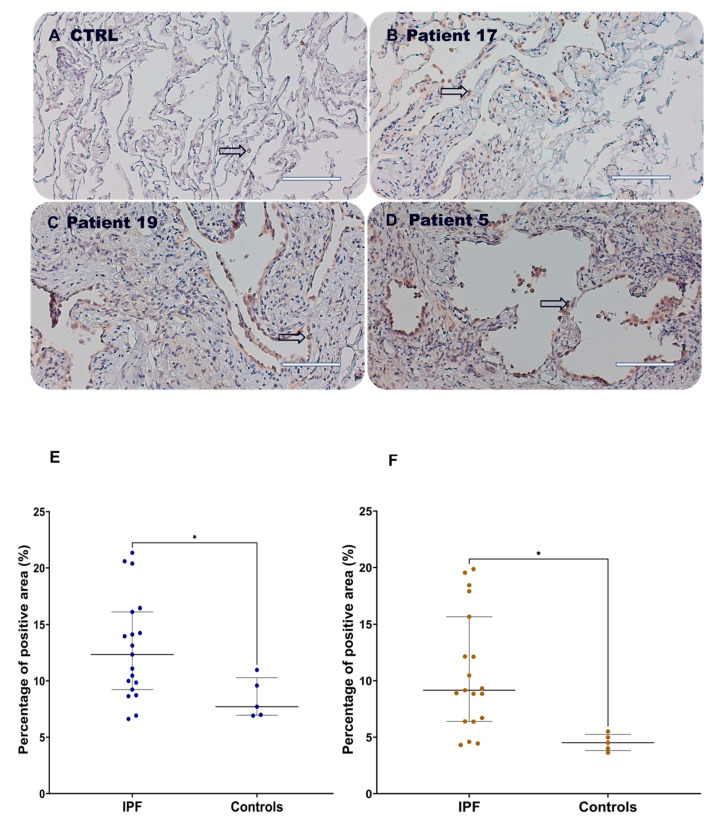
HOXB7 expression in lung parenchyma of controls and IPF patients. Representative photomicrographs of HOXB7 expression in control patient#1 (**A**), IPF patient#17 (**B**), IPF patient#19 (**C**) and IPF patient#5 (**D**) with IPF, by immunohistochemistry. Magnification 20×, scale bar 100 µm. Semi-quantitative analysis of hematoxylin (**E**) and HOXB7 (**F**) immunohistochemistry by percentage of the positive area in controls (N = 5) and patients with IPF (N = 19). Unpaired parametric *t*-test, two-tailed. (* *p*-value = 0.0454) (**E)**, (**F**) (* *p*-value = 0.0157).

**Figure 2 biomedicines-12-01321-f002:**
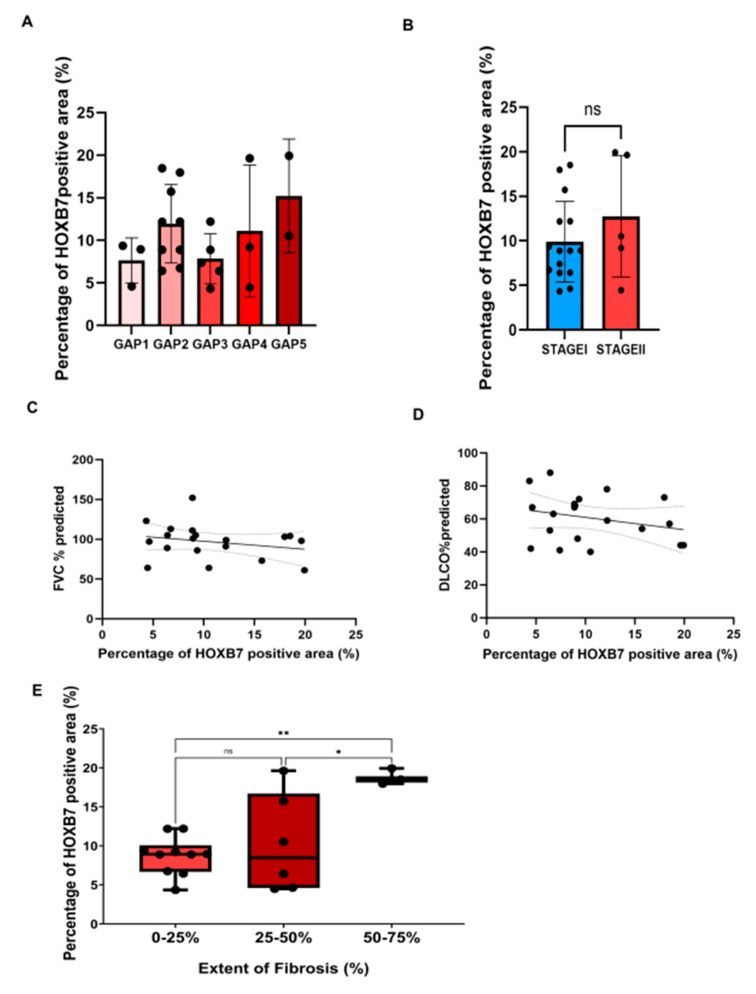
Correlations between HOXB7 expression as percentage of positive area and both lung functional decline and the extent of fibrosis: (**A**) Tukey’s multi-comparison test through ordinary one-way ANOVA (*p*-value = 0.28) showed a non-significant correlation between HOXB7 signal intensity and GAP score. (**B**) unpaired parametric *t*-test, two-tailed (*p*-value = 0.29, ns = not significant) showed non-significant difference between the percentage of HOXB7 positive area between patients at different IPF stages (stage I vs. stage II).; (**C**) nonparametric Spearman’s correlation coefficient between percentage of HOXB7 positive area and both FVC% predicted and DLCO% predicted; (**D**) showed no significant correlation between the values (r = −0.304, *p*-value = 0.196, r = −0.202, *p*-value = 0.392, respectively); (**E**) Tukey’s multi-comparisons test through ordinary one-way ANOVA showed a significant correlation between HOXB7 signal intensity and the percentage of fibrosis extent on HRCT (0–25% vs. 25–50%, mean difference −1.511 ns, 0–25% vs. 50–75%, mean difference 10.09, ** *p*-value = 0.0038, 25–50% vs. 50–75%, mean difference 9.10.09, * *p*-value: 0.0197). Error bars represent the standard error of the mean difference.

**Figure 3 biomedicines-12-01321-f003:**
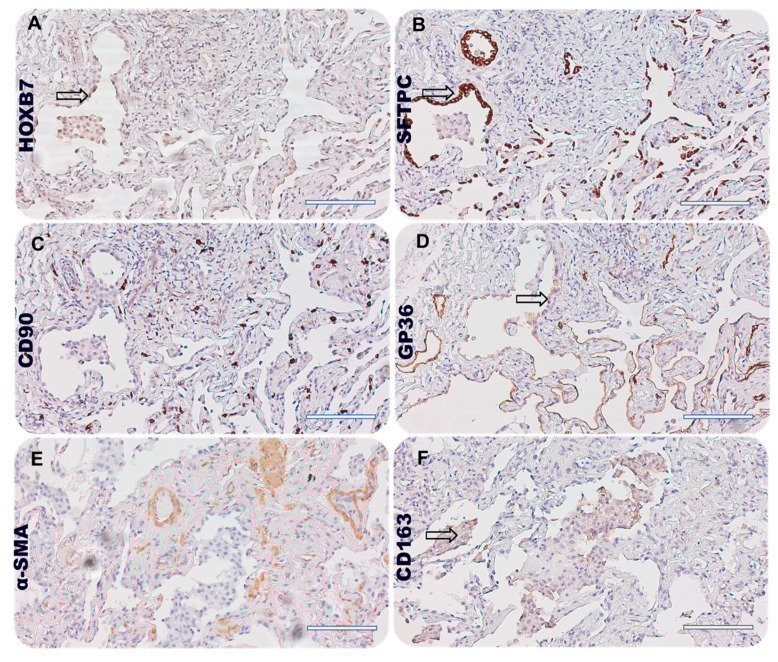
Photomicrographs showing serial immunohistochemistry (5 µm thick) of IPF patient#5, stage II, representing (**A**) HOXB7, (**B**) SFPTC, (**C**) CD90, (**D**) GP36, (**E**) α-SMA, and (**F**) CD163 staining. Magnification 20×, scale bar 100 µm, arrows represent area of partial overlapping among different staining.

**Figure 4 biomedicines-12-01321-f004:**
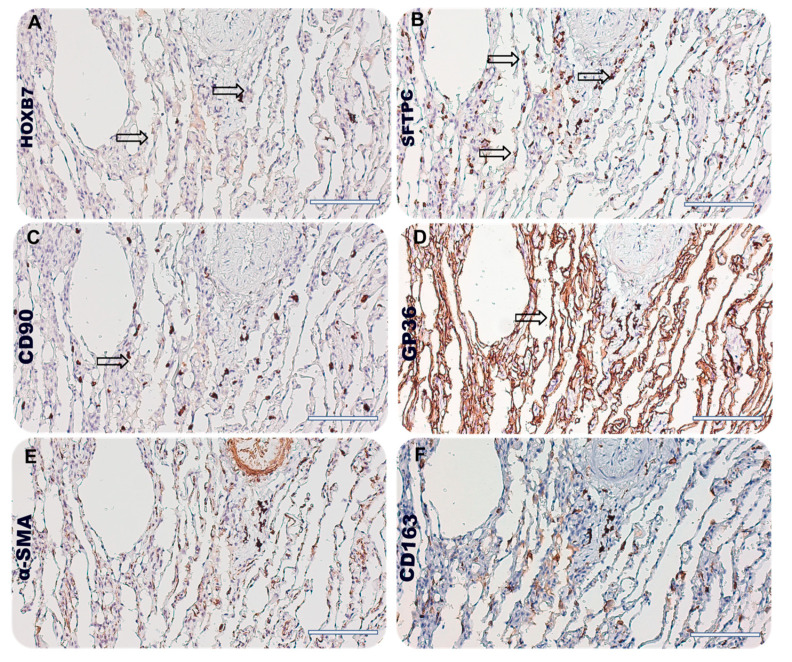
Photomicrographs showing serial immunohistochemistry (5 µm thick) of control patient#2, representing (**A**) HOXB7, (**B**) SFPTC, (**C**) CD90, (**D**) GP36, (**E**) α-SMA, and (**F**) CD163 staining. Magnification 20×, scale bar 100 µm, arrows represent area of partial overlapping among different staining.

**Table 1 biomedicines-12-01321-t001:** Characteristics of the study groups at inclusion. Data are presented as numbers (*n*) and percentages for dichotomous values or median and interquartile ranges (IQR) for continuous values.

Variable	IPF (*n* = 19)	Controls (*n* = 5)
**Age, years (IQR)**	69 (50–79)	73 (35–74)
**Male, *n* (%)**	14 (68.4)	2 (40)
**BMI, kg/m^2^ (IQR)**	28 (25–41)	32 (26–37)
**Smoking habit**		
Never, *n* (%)	5 (26.3)	2 (40)
Former, *n* (%)	12 (63.15)	3 (60)
Active, *n* (%)	3 (15.79)	0 (0)
**GAP score**		
0–3, *n* (%)	14 (73.7)	---
4–6, *n* (%)	5 (26.3)	---
**Stage**		
I, *n* (%)	14 (73.7)	---
II, *n* (%)	5 (26.3)	---
**Pulmonary function test**		
TLC, % predicted (IQR)	94 (61–108)	107 (89–110)
RV, % predicted (IQR)	98 (64–165)	132 (92–162)
FVC, % predicted (IQR)	99 (61–152)	96 (78–102)
FEV1, % predicted (IQR)	94 (67–118)	88 (65–101)
FEV1/FVC, % (IQR)	94 (60.1–98.5)	69 (68.64–85.58)
DLCO, % predicted (IQR)	63 (40–88)	62 (47–74)
**Extent of fibrosis on HRCT**		
0–25, *n* (%)	10 (53)	---
25–50, *n* (%)	6(31)	---
50–75, *n* (%)	3(16)	---

BMI = body mass index; TLC = total lung capacity; RV = residual volume; FVC = forced vital capacity; FEV1 = forced expiratory volume in 1 s; DLCO = lung diffusion test for carbon dioxide; IQR = interquartile range.

## Data Availability

The data presented in this study are available on request from the corresponding author. Therefore, data are not uploaded on a publicly available platform.

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
