# Peer review of "Expression of HOXB7 in the Lung of Patients with Idiopathic Pulmonary Fibrosis: A Proof-of-Concept Study"

_biomedicines, 2024, doi:10.3390/biomedicines12061321_

Round 1

Reviewer 1 Report

Comments and Suggestions for Authors

1. In result section 3.1 Study population, the author selected 19 patients with IPF and 5 patient with lung cancer as control. Author should justify the sample size of the study and what was the power of study.  

2. The author should plan some experiments to further validate their findings (gene expression and/or western blot). 

3. The author should justify the error bars of the statistical analysis.

4. No quantification of figure 3 and figure 4. The author should justify. 

5. Discussion is not reflecting the presented result. The author discussed overall findings of HOXB7 in the studied patients, but unable to corelate with the control subject. Author should explore the discussion with their representative images.

6. Discussion and conclusion not reflect the manuscript title, A proof of concept study and clinical significance. The author should justify the title with more corelative statements and validated data.

Author Response

On behalf of all the Authors we would like to thank all the Reviewers for their valuable time and useful contribution to our Manuscript “Expression of HOXB7 in the lung of patients with idiopathic pulmonary fibrosis: a proof-of concept study and clinical significance”. We strongly appreciate the suggestions and inputs received that will definitely improve our manuscript strengthening the rigor of the results based on the protein target HOXB7 we found highly expressed in IPF patients compared to controls. Following the Reviewers suggestions, we have reorganized the manuscript, integrated the results with new quantifications.

Comments and Suggestions for Authors

Reviewer 1

  1. In result section 3.1 Study population, the author selected 19 patients with IPF and 5 patient with lung cancer as control. Author should justify the sample size of the study and what was the power of study.

We would like to thank Reviewer 1 for the comment, whose response helped to better explain our experimental design and the cohort of patients involved in the study. The sample size was related to the explorative nature of our study aiming to investigate the expression of HOXB7 protein in fibrotic patients compared to non fibrotic controls as a proof of concept study. Then, to explain the reason of the selection of 19 patients with IPF compared to 5 controls, we have added the following in the text, in section 2.1 Study design, setting and patient identification (Page 3-4 Line 131-139):

Patients referred to the Respiratory Disease Unit of the University Hospital of Modena (Italy) who underwent SLB and/or surgical lung lobectomy within the diagnostic and therapeutic workup of lung cancer served as eligible controls considering the portion of the lung parenchyma distal to the tumor nodule and the peritumor portion. Thus, the controls were represented by portions of lung parenchyma, histologically conserved, from patients without fibrosing pathology. Then, according to these criteria, of the 20 histological samples of patients with idiopathic pulmonary fibrosis, one patient was excluded because subsequently diagnosed with pulmonary fibrosis secondary to scleroderma and not idiopathic. While of the twenty available controls only five met the eligibility criteria for this study as they had preserved and normal pulmonary alveolus architecture in comparison with the fibrotic patients.

Furthermore, following the valuable suggestion of Reviewer 1, in order to better explain the nature of the control samples, we have also modified the section Study Population 3.1, (Page 5 lines 217-222) as follow: A total of 19 patients with IPF were enrolled according to inclusion criteria while 5 patients with lung cancer diagnosis served as controls since the normal lung parenchima far from lung cancer, without fibrosing pattern, was analyzed.

  1. The author should plan some experiments to further validate their findings (gene expression and/or western blot).

We agree with the Reviewer, but unfortunately we were not able to perform the Western Blot analysis since  we not have enough paraffin material available for the experiment. Most of the material was used to perform our retrospective study in which we developed a protocol to optimise protein yield from FFPE [1]. However, we aim to perform this experiment on fresh biopsies from IPF patients compared to controls, to test the expression of HOXB7 in a prospective study model that is currenlty ongoing in our Research group.

  1. 3. The author should justify the error bars of the statistical analysis.

We thank the Reviewer for pointing out this issue in our statystical analysis. The error bars represent the standard error of mean difference from Tukey‘s multiple comparisons test through Ordinary One-way Anova as explained in the legend of Figure 2A and Figure 2E, while in the bargraph of Figure 2B, the error bars show the standard error of mean (SEM), though the unpaired t-test (two tailed). Taking into account the suggestions of Reviewer 1, we have implemented the Legend in Figure 2 (page 11, lines 280-294) with the above-mentioned comments.   

  1. No quantification of figure 3 and figure 4. The author should justify.

    We thank the reviewer for highlighting this point and we performed the semi-quantitative analysis on both Figure Panel 3 and Figure Panel 4, showing the lung niche marker‘s positive area in Supplementary   Figure 3. Then, we have modified within the text as followed (Page 12, Line 321-328): Then, the semi-quantitative analysis of the stained area for HOXB7 and the markers of the lung niche of IPF patient, showed that HOXB7 stained area was higher in IPF patients compared to controls patients, while the GP-36 stained area was increased in normal lung parenchyma probably due to the pathobiology of idiopathic pulmonary fibrosis involving AECI depletion (Supplementary Figure 3). Then, the expression level of both SFTPC and CD163 were similar between the IPF patients and controls, while α-SMA was highly expressed in patients with IPF compared to the controls.

  1. Discussion is not reflecting the presented result. The author discussed overall findings of HOXB7 in the studied patients, but unable to correlate with the control subject. Author should explore the discussion with their representative images.

We agree with the Reviewer and we tried to better explain our findings in the Discussion section comparing the fibrotic patients with the controls. We have revised the Discussion considering the implementation of the results with the data deriving from the semi-quantitative analysis of the serial section: (Page 15 Line 348-351). To the best of our knowledge this is a first study that provides evidence that HOXB7 protein expression is higher in the lung of patients with IPF compared with the lung of patients with no signs of lung fibrosis that represented our controls. Page 16 Line 374-384: The serial IHC analysis with the specific antibodies of the lung niche (Figure 3, Figure 4), showed that SFTPC-positive AECII distribute differently in the pulmonary alveolus of the fibrotic patient(Figure 3B) compared to the controls (Figure 4B), occupying the lumen of the pulmonary interstitium, which is normally covered by AECI. It was  in this histological region characterised by AECII that

were probably not able to transdifferentiate into AECI, according to the pathobiogenesis of the IPF causing the depletion of AECI and potential AECII hyperplasia, that we observed the most similarity with the staining areas positive for HOXB7.Thus, we can speculate that HOXB7 may contribute to the epithelial barrier dysfunction, involving the depletion of AECI and the hyperplasia of AECII together with the action of proinflammatory cytokines TGF-βTNF-α, IL-1β and reactive oxygen species (ROS) [36].

6.Discussion and conclusion not reflect the manuscript title, A proof of concept study and clinical significance. The author should justify the title with more corelative statements and validated data.

We agree with the pertinent suggestion of the Reviewer 1 and we have changed the Manuscript title accordingly: Expression of HOXB7 in the lung of patients with idiopathic pulmonary fibrosis: a proof of concept study which is more related to the findings of our project.

Reviewer 2 Report

Comments and Suggestions for Authors

This manuscript describes the expression of HOXB7 in lung tissue of patients with idiopathic pulmonary fibrosis; the authors also attempt to demonstrate the correlation between HOXB7 expression and fibrosis severity; the study provides interesting findings; however, some aspects need improvement. Listed below are some issues that need to be addressed:

1.         In the introduction the authors highlight the involvement of HOXB7 in different types of cancer. However, they decided to use tissues from lung cancer patients as controls. Please justify the choice of the control group.

2.         The authors report hematoxylin quantification, but it's not clear what the purpose of this quantification is. It would be beneficial for the authors to explain what information they obtained from this quantification and its significance in the context of their study.

3.         Additionally, the authors quantified HOXB7 expression from IHC and reported its expression as average pixel intensity. It is important to keep in mind that the use of the Color Deconvolution 2 plugin by ImageJ is not recommended to quantify intensity since immunohistochemical methods are not stoichiometric, so it is not recommended to measure expression as a function of DAB intensity.

It is suggested that data be reported by area or percentage of positive area for each marker or by the number of positive nuclei.

4.         Figures 3 and 4 are the same. In the caption of Figure 4, patient #2 is indicated to be presented. Please verify and add the corresponding figures.

5.         The authors mention that they performed serial sections to evaluate different molecules by IHC and demonstrate a colocalization in the expression of these. However, it's important to note that this may not be the most appropriate method for this purpose. Alternative methods, such as double labeling by immunofluorescence, should be considered for a more accurate evaluation of colocalization.

6.         Please verify that your references match the text. For example, reference 24 does not mention anything cited in the text.

Author Response

Reviewer 2

Comments and Suggestions for Authors

This manuscript describes the expression of HOXB7 in lung tissue of patients with idiopathic pulmonary fibrosis; the authors also attempt to demonstrate the correlation between HOXB7 expression and fibrosis severity; the study provides interesting findings; however, some aspects need improvement. Listed below are some issues that need to be addressed:

  1. In the introduction the authors highlight the involvement of HOXB7 in different types of cancer. However, they decided to use tissues from lung cancer patients as controls. Please justify the choice of the control group.

We thank the Reviewer for the comment and apologise for not having described the nature of our control samples in detail and the need of their enrollment in this project.  We have used the portion of normal (or conserved) lung parenchyma distal from lung cancer as control specimens because it is difficult or at least very rare, to find biopsies of healthy lungs also referred to as non-disease control. The availability of a healthy non-diseased control lung biopsy, comes from donors whose lungs were deemed unsuitable for transplantation, which is a very low probability for our country. Thus, we have used the conserved lung parenchyma far from lung cancer noduls (where HOXB7 is under expressed compared to either fibrotic parenchyma or tumor stroma) as control and non fbrotic parenchyma.

Furthermore, we have added the following within the Manuscript in response to the pertinent comment of both Reviewer 2 and previuos Reviewer 1 who also rightly raised this issue. Section 2.1 Study design, setting and patient identification  (Page 3-4 Line 131-139):

Patients referred to the Respiratory Disease Unit of the University Hospital of Modena (Italy) who underwent SLB and/or surgical lung lobectomy within the diagnostic and therapeutic workup of lung cancer served as

eligible controls considering the portion of the lung parenchyma distal to the tumor nodule and the peritumor portion. Thus, the controls were represented by portions of lung parenchyma, histologically conserved, from patients without fibrosing pathology. Then, according to these criteria, of the 20 histological samples of patients with idiopathic pulmonary fibrosis, one patient was excluded because subsequently diagnosed with pulmonary fibrosis secondary to scleroderma and not idiopathic. While of the twenty available controls only five met the eligibility criteria for this study as they had preserved and normal pulmonary alveolus architecture in comparison with the fibrotic patients.

   Furthermore, following the valuable suggestion of Reviewer 2, in order to better explain the nature of the control samples, the subject of this study, we have also corrected the section Study Population 3.1, (Page 5 lines 217-222): A total of 19 patients with IPF were enrolled according to inclusion criteria while 5 patients with lung cancer diagnosis served as controls since the normal lung parenchima far from lung cancer, without fibrosing pattern, was analyzed.

  1. The authors report hematoxylin quantification, but it's not clear what the purpose of this quantification is. It would be beneficial for the authors to explain what information they obtained from this quantification and its significance in the context of their study.

We thank the Reviewer 2 for the comment whose response allowed us to explain the data in more detail. The quantification of the haematoxylin was done to check whether the expression profile of haematoxylin and thus (indirectly) the number of cells was comparable to the expression profile of HOXB7, from the semi-quantitative analysis. This allowed to check whether there was a correlation between them as shown in the correlation graph in Supplementary Figure 1, in IPF patients. Then, thaks to the valuable comment of the Reviewer 2 we thought to quantify also the non-linear correlation between  haematoxylin signal and the HOXB7 expression in control patients, by adding a Supplementary Figure 2 within the Manuscript. Thus, we added the following details in the Manuscript to answer to the Reviewer 2 (Page 6- 7, Line 244-252) providing the significance of this results obtained from the analysis: A medium positive linear correlation was found between the HOXB7 protein expression and hematoxylin (r= 0.53 95%CI [0.116 – 1.114], p=0.0158, Supplemental Figure 1S) in the lung parenchyma of patients with IPF suggesting partial localization of HOXB7 protein in the nuclear compartment of proliferating cells during the progression of the disease. The weak linear correlation between the HOXB7 protein expression and hematoxylin (r<0.1 p=0.956, ns, Supplemental Figure 2S) in control patients, suggests that in the lung parenchyma of control patients, HOXB7 is not primarily present in the nucleus of normal lung alveolar cells since an increase in the number of nuclei and thus cells does not always correspond to an increase in HOXB7 expression. This result shows that HOXB7 is distributed in different cellular compartments in the lung parenchyma of fibrotic patients.

  1. Additionally, the authors quantified HOXB7 expression from IHC and reported its expression as average pixel intensity. It is important to keep in mind that the use of the Color Deconvolution 2 plugin by ImageJ is not recommended to quantify intensity since immunohistochemical methods are not stoichiometric, so it is not recommended to measure expression as a function of DAB intensity. It is suggested that data be reported by area or percentage of positive area for each marker or by the

number of positive nuclei. We thank the reviewer for the relevant comment, as the semi-quantitative analysis of immunohistochemistry using the Deconvolution 2 plugin cannot be considered stoichiometric. Thus, we have modified all graphs showing the expression of protein markers by reporting the percentage of positive staining area, relative to the antibody used, instead of the average pixel intensity (Figures 1E and 1F, Figures 2A, B, C, D and E, Supplemental Figure 3). Consequently, we changed the word quantification in the text and replaced it with a semi-quantitative analysis due to its non-steichiometric nature, expressed as a percentage of the positive area.

  1. Figures 3 and 4 are the same. In the caption of Figure 4, patient #2 is indicated to be presented. Please verify and add the corresponding figures.

We thank the reviewer for identifying the mistake and we have placed Figure Panel 4 in the text, accordingly.

  1. The authors mention that they performed serial sections to evaluate different molecules by IHC and demonstrate a colocalization in the expression of these. However, it's important to note that this may not be the most appropriate method for this purpose. Alternative methods, such as double labeling by immunofluorescence, should be considered for a more accurate evaluation of colocalization.

We thank the reviewer and apologise for misusing the term colocalisation, which presupposes, as the reviewer rightly suggested, the use of double immunohistochemistry or immunofluorescence. Our intention was to compare the expression profile of HOXB7 with that of the other lung markers, so also in response to Reviewer 1 we compared the positive area percentage for HOXB7 with the positive area percentage of the other markers both in fibrotic lung parenchyma and the control (Supplemental Figure 3). As we explain in the text, Page 12 Line 318-330 Discussion Line 376-383, the qualitative evaluation of the IHC serial slides shows us that HOXB7 appears to share histological regions with both alveolar type II cells in a potential  hyperplasia and type I cells that are present to a lesser extent due to their depletion during the pathobiogenesis of IPF.

Discussion Line 318-330 The serial IHC analysis with the specific antibodies of the lung niche (Figure 3, Figure 4), showed that SFTPC-positive AECII distribute differently in the pulmonary alveolus of the fibrotic patient(Figure 3B) compared to the controls (Figure 4B), occupying the lumen of the pulmonary interstitium, which is normally covered by AECI. It was  in this histological region characterized by AECII that were probably not able to transdifferentiate into AECI, according to the pathobiogenesis of the IPF causing the depletion of AECI and potential AECII hyperplasia, that we observed the most similarity with the staining areas positive for HOXB7

  1. Please verify that your references match the text. For example, reference 24 does not mention anything cited in the text.

We thank the Reviewer 2 and we placed reference 24 in the correct position

Reviewer 3 Report

Comments and Suggestions for Authors

The authors investigated the molecular pathways involved in the initiation and progression of idiopathic pulmonary fibrosis (IPF), some of which are shared with the development of lung cancer. HOXB7, a member of the homeobox (HOX) gene family, has been found to be involved in various cancers. The authors offer a thorough and statistically sound study. The proposed findings will support the notion of HOXB7- expression being higher in the lungs of IPF patients compared to controls, being represented in different cellular compartments in the lung niche.

Comment:

1. The presented tables are boring. The description of the indicators has been moved. I suggest making them colorful and more detailed.

2. The discussion presented is too small and insufficient. More detailed justification to be added.

3. The conclussion part have to b reviced. 

Comments on the Quality of English Language

 Minor editing of English language required

Author Response

Reviewer 3

Comments and Suggestions for Authors

The authors investigated the molecular pathways involved in the initiation and progression of idiopathic pulmonary fibrosis (IPF), some of which are shared with the development of lung cancer. HOXB7, a member of the homeobox (HOX) gene family, has been found to be involved in various cancers. The authors offer a thorough and statistically sound study. The proposed findings will support the notion of HOXB7- expression being higher in the lungs of IPF patients compared to controls, being represented in different cellular compartments in the lung niche.

Comment:

  1. The presented tables are boring. The description of the indicators has been moved. I suggest making them colorful and more detailed.

We thank the Reviewer 3 and we modified the table by adjusting the layout and using greyscale colors.

  1. The discussion presented is too small and insufficient. More detailed justification to be added.

We thank the Reviewer 3 and we implemented the Discussion with the potential role of increased HOXB7 in the loss of function of the pulmonary alveolar epithelium in IPF patients. Page 15-16 Line 350-398:

To the best of our knowledge this is a first study that provides evidence that HOXB7 protein expression is higher in the lung of patients with IPF compared with the lung of patients with no signs of lung fibrosis that represented our controls. Thus far, very few studies have explored the expression of HOXB7 in the lung disease, and none of them has focused onto the onset and development of lung fibrosis [31], [32], [33]. Here, we found a significant expression of HOXB7 in the lung parenchyma of IPF patients based on IHC analysis;. Despite the relatively small cohort of patients with IPF, our study population was highly representative of the IPF patient population in terms of clinical and functional features. 

HOXB7 per se is known to contribute to cancer progression, promoting epithelial-to-mesenchymal transition [34], anticancer drug resistance and angiogenesis. As such, the amplification and the overexpression of HOXB7 have been found to correlate with poor prognosis in various cancers such as gastric [35], pancreatic [13] and lung cancer [31]. In our series of patients with IPF, we have observed that HOXB7 does not appear to correlate with increased clinical severity of patients with IPF in terms of lung function decline (FVC and DLCO). However, we have quite clearly shown that HOXB7 expression may be correlated with the disease extension as evident at the lung scan imaging. 

It might be possible that lack of correlation between HOXB7 expression and the lung function derived from the low number of patients included in the analysis and the clinical stage subgroup representation (see in Table 1). Nonetheless, it may be possible that the higher cellular proliferation and representation in IPF, mirrored by higher amount of cell nuclei, explain results on the HOXB7 positivity. This in turn might drive fibrotic progression leading to a greater extent in the lung parenchyma of patients with IPF compared with controls. Given these premises a potential association between HOXB7 and IPF severity/progression might be

therefore hypothesized.  The serial IHC analysis with the specific antibodies of the lung niche (Figure 3, Figure 4), showed that SFTPC-positive AECII distribute differently in the pulmonary alveolus of the fibrotic patient (Figure 3B) compared to the controls (Figure 4B), occupying the lumen of the pulmonary interstitium, which is normally covered by AECI. It was  in this histological region characterised by AECII that were probably not able to transdifferentiate into AECI, according to the pathobiogenesis of the IPF causing the depletion of AECI and potential AECII hyperplasia, that we observed the most similarity with the staining areas positive for HOXB7.Thus, we can speculate that HOXB7 may contribute to the epithelial barrier dysfunction, involving the depletion of AECI and the hyperplasia of AECII together with the action of proinflammatory cytokines TGF-βTNF-α, IL-1β and reactive oxygen species (ROS)[36]. Furthermore, a potential expression of HOXB7 in alveolar macrophages (AM) could contribute to the existing cross-talk between alveolar macrophages and lung fibroblasts. In particular, it has been found that macrophages might influence the gene expression of fibroblasts according to their activation state, increasing the expression of collagens, αSMA and TGFβ [37], as well as ECM synthesis and fibroblast proliferation [38]. As such, this cellular involvement and biological action driven by HOXB7 it is likely to promote the profibrotic pathway which is the hallmark of IPF.

  1. The conclussion part have to b reviced. 

We thank the Reviewer 3 and we revised the Conclusions of our Manuscript (Page 16-17 Line 413-424): Given its main limitation, e.g. the small number of patients included, the present study was able to show that HOXB7, a member of the homeobox gene family involved in regulating the proliferation, motility, and angiogenesis of various solid and non-solid neoplastic diseases, is highly represented in the lung tissue of patients with IPF, and it seems to be associated with the extension of disease within this organ. We were also able to demonstrate that HOXB7 might be involved both in the proliferative phase of fibrotic disease and in the alveolar epithelium dysfunction since its histological expression profile was associated with an aberrant distribution of alveolar type II and a depletion of alveolar type I cells, associated with the pathogenesis of IPF. Therefore, it is likely that HOXB7 may act as a biomarker of disease progression.

Round 2

Reviewer 3 Report

Comments and Suggestions for Authors

-

Author Response

Dear Editor,

We sincerely appreciate your response and the positive evaluation of our revision. We acknowledge your observation. However, our project adheres to a protocol approved by our ethics board, which pre-specified and approved the number of controls as reported in the manuscript.
Additionally, the project was funded based on the number of patients declared in the protocol.

Therefore, it is currently not feasible for us to comply with your request, despite its legitimacy, as it would require an extended period to obtain new approval from the ethics board and secure additional funding for further analyses.

Thank you for your understanding.

Yours sincerely,

Anna Valeria Samarelli, PhD
Roberto Tonelli, MD, PhD
prof. Enrico Clini, MD, FERS, FCCP